# The Mechanism of miR-21-5p/TSP-1-Mediating Exercise on the Function of Endothelial Progenitor Cells in Aged Rats

**DOI:** 10.3390/ijerph20021255

**Published:** 2023-01-10

**Authors:** Xiaoke Chen, Kejia Xie, Xinzheng Sun, Chengzhu Zhang, Hui He

**Affiliations:** 1School of Sports Science, Beijing Sport University, Beijing 100084, China; 2China Institute of Sport and Health Science, Beijing Sport University, Beijing 100084, China

**Keywords:** aerobic exercise, endothelial progenitor cells, miR-21-5p, thrombospondin 1

## Abstract

(1) Background: The declined function of peripheral circulating endothelial progenitor cells (EPCs) in aging individuals resulted in decreased endothelial cell regeneration and vascular endothelial function. Improving EPCs function in aging individuals plays an important role in preventing cardiovascular diseases. (2) Methods: Thirty aged (18-month-old) male Sprague-Dawley rats were randomly divided into control and exercise groups. An aerobic exercise intervention was performed 5 days/week for 8 weeks. EPCs functions, miR-21-5p, and TSP-1 expressions were detected after the intervention. The senescence rate, proliferation, and migration of EPCs were examined after overexpression of miR-21-5p and inhibition of TSP-1 expression. (3) Results: The senescence rate, proliferation, and migration of EPCs in exercise groups were significantly improved after exercise intervention. The miR-21-5p expression was increased and the TSP-1 mRNA expression was decreased in the EPCs after the intervention. miR-21-5p overexpression can improve EPCs function and inhibit TSP-1 expression but has no effect on senescence rate. Inhibition of TSP-1 expression could improve the function and reduce the senescence rate. (4) Conclusions: Our results indicate that long-term aerobic exercise can improve the functions of EPCs in aging individuals by downregulating TSP-1 expression via miR-21-5p, which reveals the mechanism of exercise in improving cardiovascular function.

## 1. Introduction

Aging leads to a decline in vascular endothelial function, which induces intimal hyperplasia and causes atherosclerosis and other cardiovascular diseases [1]. Endothelial cells (ECs) can regulate the contraction and relaxation of blood vessels by releasing vasoactive substances. Once ECs are damaged, various pathological changes, such as atherosclerosis, vascular stenosis, and blockage, can easily occur [2]. Owing to the limited and slow self-proliferation and repair ability of ECs after injury, endothelial progenitor cells (EPCs) in bone marrow and blood are mobilized and induced to differentiate into ECs to repair damaged blood vessels [3]. A previous study found that aging resulted in a decrease in the basic level of EPCs in peripheral circulation, significant reduction of EPCs functions, such as migration and proliferation, and deterioration in the ability of EPCs to repair and renew damaged blood vessels [4].

Studies have confirmed that exercise can promote the mobilization of EPCs from bone marrow to the site of damaged vascular endothelial repair. Acute exercise can promote the mobilization of EPCs into the peripheral circulation and increase the peripheral circulating content [5]. Studies have found that the function of circulating EPCs can be significantly improved after exercise for at least 8 weeks in elderly adults, obese children, and people with cardiovascular diseases [5,6,7].

Reports on the mechanisms of the mobilization and improvement of function of EPCs have gradually increased. As microRNA (miRNA) plays an important role in the regulation of gene expression, more attention has been paid in recent years to the effect of changes in miRNA expression in EPCs regulating the signaling pathway on cell function [8,9,10]. miR-21-5p plays an important regulatory role in protecting cardiac myocytes and the vascular endothelium. Mauryan et al. [11] found that exosomes secreted by human bone marrow mesenchymal stem cells (BMSCs) are rich in miR-21-5p, which can effectively enhance the contractility of myocardial tissue. Exercise has been found to increase the expression levels of miR-21-5p in EPCs in mice with heart failure [12], which suggests that exercise may improve cell function by affecting miR-21-5p expression in EPCs.

Thrombospondin 1 (TSP-1) plays an important role in aging cells. Previous studies have found that aging factors can lead to the upregulation of PTS-1 expression, especially in cardiomyocytes [13,14], resulting in decreased cell function and accelerated aging. Upregulation of TSP-1 expression may lead to decreased EPCs function and angiogenesis ability [15,16]. However, few studies support this theory, and further studies are needed to confirm whether the upregulation of TPS-1 expression caused by aging is a cause of the decline of EPCs function. Moreover, whether exercise can improve cell function and senescence by decreasing TPS-1 expression levels remains to be investigated.

In onclusion, previous studies have found that exercise can improve the function of EPCs, but the mechanism is still unclear. Hu et al. [17] found that EPCs transported miR-21-5p into ECs through exocrine and negatively regulated the expression of TSP-1 to improve the function of ECs. Therefore, we speculated that the miR-21-5p/TSP-1 pathway may be involved in the regulation of EPCs function by exercise. In this study, the function of EPCs and the expression levels of miR-21-5p and TSP-1 in aging rats were observed after 8 weeks of aerobic exercise. The senescence rate and proliferation and migration functions of EPCs were observed by regulating the expressions of miR-21-5p and TSP-1 to examine whether exercise can improve the function of EPCs in aging individuals and further investigate the molecular mechanism of exercise to improve EPCs function. The significance of this study is to further reveal the mechanism of exercise improving the function of EPCs and the interaction between genes so as to lay a theoretical foundation for the experimental study of exercise improving vascular function in the future and provide scientific evidence for exercise to promote cardiovascular health.

## 2. Materials and Methods

### 2.1. Subjects and Exercise Intervention

Thirty aged (18-month-old) male Sprague-Dawley (SD) rats were divided into control (CG) and exercise (EG) groups, with 15 rats in each group. The rats were purchased from the Beijing Keyu Animal Breeding Center. This experiment was approved by the Sports Science Experiment Ethics Committee of Beijing Sport University (approval number: 2020149A). All animal experiments in this study were carried out in accordance with the principles of management and the use of local laboratory animals and followed the guidelines for the management and use of laboratory animals published by the National Institutes of health.

Aerobic exercise intervention was performed for 8 weeks, 5 days per week. The exercise intervention program is shown in Table 1 [18]. Before the formal experimental exercise, the rats were acclimated to the treadmill for 4 days at an acclimation intensity of 5 m/min × 10 min on the first and second days and at 10 m/min ×15 min on the third and fourth days. The rats were allowed to eat and drink freely during the exercise intervention.

### 2.2. Cell Culture and Identification

EPCs were isolated and cultured as previously described [19]. In brief, peripheral blood mononuclear cells (PBMNCs) from the rats in each group were isolated by Ficoll density gradient centrifugation after the 8-week exercise intervention. Blood samples were collected 48 h after exercise cessation. After washing with phosphate-buffered saline (PBS) 3 times, the PBMNCs were resuspended in EC basal medium 2 (EGM-2MV; CC-3202, Lonza, Basel, Switzerland) containing 10% fetal bovine serum and inoculated in 6-well culture plates coated with fibronectin. After 3 days in culture, nonadherent cells were abandoned. Adherent cells were continuously cultured for subsequent experiments. Then, the medium was changed every 2 days. EPCs were identified on the basis of their typical spindle-shaped morphology.

EPCs were confirmed by assessing acetylated low-density lipoprotein (ac-LDL) uptake and ulex europaeus agglutinin-1 (UEA-l) binding. Cells were incubated with Dil-ac-LDL (15 μg/mL) for 4 h at 37 °C and fixed with 4% paraformaldehyde for 20 min. After washing with PBS 3 times, the cells were stained with FITC-UEA-l (10 μg/mL) for 1 h at 37 °C. Finally, the cells were analyzed using a laser confocal fluorescence microscope (Leica Microsystems, Wetzlar, Germany).

### 2.3. Senescence-Associated Beta-Galactosidase Staining

EPC senescence was determined using a senescence-associated beta-galactosidase (SA-β-gal) staining kit (Beyotime, Haimen, China). EPCs were fixed with a fixation buffer at room temperature for 15 min. After washing with PBS 3 times, the cells were incubated with a working solution at 37 °C without CO_2_ for 12 h. PBS was added to stop the reaction. The ratio of the β-gal-positive cells to the total number of cells was determined under light microscopy.

### 2.4. Proliferation Assay

EPC proliferation was determined with a 3-(4,5-dimethylthiazol-2-yl)-2,5-diphenyltetrazolium bromide (MTT) assay, in accordance with the manufacturer’s protocol. The cell concentration was adjusted to 5 × 10^4^ cells/mL, and MTT-tetrazolium salt (Beyotime) was added at the same time for the first 3 days. After 4 h of incubation, formazan crystals were dissolved by adding 2-propanol. The absorbance (optical density (OD)) value was measured at 490 nm. The growth curve of the EPC cells was plotted using the culture time as the abscissa and the average OD value in each group as the ordinate.

### 2.5. Migration Assay

EPCs were plated (at 5 × 10^4^ cells per well) in 6-well plates and incubated at 37 °C for 24 h in serum-free EGM media. When the cell confluence reached 80–90%, the cells were scratched with a 200 µL pipette tip. The debris were washed off with PBS, and the cells were cultured with a serum-free medium at 37 °C. Cell migration was evaluated at 0 and 24 h after scratching. The migration area was calculated as follows: migration area (%) = (A0 – An)/A0 × 100, where A0 is the initial wound area, and An is the wound residual area at the measurement point.

### 2.6. Cell Transfection

To regulate the expressions of miR-21-5p and TSP-1 in EPCs, miR-21-5p mimic, mimic-negative control (NC), TSP-1 siRNAs, and siRNA control were transfected into cells with Lipofectamine 3000, in accordance with the manufacturer’s instructions (GenePharma, Shanghai, China). The primer sequences are shown in Table 2. After 72 h of incubation, the EPCs were harvested for subsequent experiments.

### 2.7. Quantitative Real-Time Polymerase Chain Reaction Analysis

Total RNA was extracted with a Trizol reagent (Invitrogen, Carlsbad, CA, USA). A cDNA reverse transcription kit (Invitrogen) was used to perform reverse transcription. Mature miRNA expression was analyzed using a miRNA quantitative real-time polymerase chain reaction (qRT-PCR) kit (Applied Biosystems, Foster City, CA, USA). U6 and β-actin were used as internal controls. Relative gene expressions were later quantified using the 2^−ΔΔCT^ calculation method.

### 2.8. Western Blot

Total proteins were extracted from the EPCs using a radioimmunoprecipitation assay buffer (Beyotime), and the protein concentration was measured using a bicinchoninic acid protein assay kit (Beyotime). Total protein was isolated using sodium dodecyl sulfate-polyacrylamide gel and transferred to a polyvinylidene difluoride membrane. Western blotting analysis was probed using a primary anti-TSP-1 antibody (Sigma-Aldrich, St. Louis, MO, USA). Tubulin level (Sigma) was measured as the internal control.

### 2.9. Statistical Analysis

Data are shown as mean ± standard deviation (SD). An unpaired *t* test was used for comparing two independent groups. Comparisons of multiple independent groups were analyzed using a one-way analysis of variance (ANOVA). Changes in cell proliferation were analyzed using a two-way ANOVA. The GraphPad Prism software (version 8.3.0, GraphPad Software, San Diego, CA, USA) was used for statistical analysis. A *p*-value of <0.05 was considered to denote statistical significance.

## 3. Results

### 3.1. Identification of EPCs

At the end of the 8-week exercise intervention, blood samples for the separation of EPCs were extracted from the EG and CG. The mononuclear cells isolated on the first day were small and round in shape. After 2 weeks of differentiation culture, the EPCs were enlarged and gradually elongated in a spindle shape, germinating from the center of the colony. The cells exhibited the ability to absorb ac-LDL and bind to endothelium-specific lectin UEA-1 (Figure 1), which is characteristic of EPCs.

### 3.2. Effect of 8-Week Aerobic Exercise on the Senescence, Proliferation, and Migration of EPCs in Aged Rats

After the 8-week aerobic exercise intervention, EPCs in the peripheral blood samples from the rats in the EG and CG were isolated and cultured to determine their aging rate and proliferation and migration abilities. SA-β-gal staining revealed that the senescence rate of the EPCs in the EG was significantly lower than that of the EPCs in the CG (Figure 2a). The MTT assay revealed that the proliferation rate of EPCs was higher in the EG than in the CG at 24 and 48 h (Figure 2b). The migration rate in the EG was significantly better than that in the CG 24 h after scratching (Figure 2c). qRT-PCR revealed that the expression of miR-21-5p in peripheral EPCs after 8 weeks of intervention was significantly higher in the EG than in the CG, whereas the expression of TSP-1 mRNA was significantly lower in the CG (Figure 2d). We speculated that exercise may improve the function of EPCs by upregulating the expression of miR-21-5p and downregulating the expression of TSP-1 in aging individuals.

### 3.3. Effects of miR-21-5p Overexpression on EPCs Function and TSP-1 Expression

To verify the above-mentioned conjecture, miR-21-5p overexpression was induced to determine the mechanism by which exercise improves EPCs function in aging individuals. After transfection with miR-21-5p mimic into EPCs derived from the peripheral blood of elderly rats, the effects of miR-21-5p overexpression on the senescence, proliferation, and migration of and TSP-1 expression in EPCs were investigated. qRT-PCR revealed that the miR-21-5p expression in EPCs was significantly increased after transfection with miR-21-5p mimic for 24 h (Figure 3a). No significant difference in cell senescence rate after transfection was found among all the groups (Figure 3b), indicating that upregulation of miR-21-5p expression may have no effect on cell senescence. The proliferation and migration functions of EPCs were significantly improved in the miR-21-5p mimic group compared with the NC and blank groups, with no significant difference between the NC and blank groups (Figure 3c,d). The expression of TSP-1 protein was significantly lower in the miR-21-5p mimic group than in the NC and blank groups, with no significant difference between the NC and blank groups (Figure 3e), indicating that miR-21-5p is a key regulator of the proliferation and migration functions of EPCs via TSP-1 expression.

### 3.4. Effects of TSP-1 Expression on EPCs Function

To explore the effect of TSP-1 expression on EPCs function, TSP-1 siRNA was transfected into EPCs derived from the peripheral blood of elderly rats, and cell senescence rate and functional changes were detected 24 h after transfection. The results showed that the senescence rate and proliferation and migration functions of the EPCs in the TSP-1 siRNA group were significantly improved compared with those in the control and blank groups, with no significant differences between the control and blank groups (Figure 4a–c).

## 4. Discussion

In this study, we found that long-term aerobic exercise can improve the proliferation and migration functions of EPCs in aging individuals via upregulating miR-21-5p expression and inhibiting TSP-1 expression and can reduce the EPCs senescence rate by downregulating TSP-1 expression. The results of molecular experiments emphasized that miR-21-5p is a key regulator of the proliferation and migration functions of EPCs via TSP-1 (Figure 5).

Cell senescence refers to the loss of cell proliferation ability and the permanent stagnation of the cell cycle. The abilities of aging EPCs to proliferate, migrate, and directionally differentiate into ECs to repair blood vessels are reduced [20]. Delaying the senescence of EPCs and extending their physiological functions can help maintain the normal storage content and function of peripheral blood cells. In this study, long-term aerobic exercise can reduce the senescence rate of peripheral blood EPCs in elderly rats, for the following reasons: studies have shown that long-term exercise effectively delayed the aging process of cells and that TSP-1 expression, which is involved in the aging process of cells and can promote the aging of ECs or cardiomyocytes to a certain extent [13,14], was downregulated, indicating that exercise may delay cell senescence by downregulating the TSP-1 expression in EPCs. In addition, exercise may reduce the rate of aging by promoting the mobilization of bone-marrow-derived EPCs into the bloodstream to increase the proportion of new cells.

The mobilization and differentiation abilities of bone-marrow-derived cells decline in aging individuals, resulting in impaired proliferation, migration, and angiogenesis of peripheral EPCs [4]. Improving the function of peripheral EPCs in aging individuals is crucial for the prevention of cardiovascular diseases. In this study, we found that long-term aerobic exercise can improve the proliferation and migration functions of EPCs in aging individuals, consistent with previous findings [4,20]. Some studies have also found that exercise can improve the ability of EPCs to locate to the damaged sites of blood vessels and differentiate into ECs for repair [20]. Improving the function of EPCs can help cells quickly locate to the damaged parts of blood vessels for rapid proliferation and repair of blood vessels, which are of great significance in preventing the decline of vascular endothelial function in aging individuals and the development of cardiovascular diseases.

Previous studies have found that long-term exercise can promote the increase of miR-21-5p expression. Souza et al. [12] detected changes in the expression levels of 53 miRNAs related to myocardial regulation in rat cardiomyocytes after 10 weeks of aerobic exercise intervention and found that long-term exercise can significantly upregulate miR-21-5p expression. A significant increase in plasma circulating miR-21-5p was also found in long-term synchronized swimming athletes [21]. Another study found that acute exercise can also promote an increase in miR-21-5p expression level in cells [22].

miR-21-5p has been proven to be related to the angiogenesis and repair and to be involved in the generation and regulation of vascular ECs [17,23,24]. Upregulation of miR-21-5p expression can improve the function and angiogenesis of vascular ECs [17,23,25]. Ge et al. [23] found that miR-21-5p expression was upregulated in rats with traumatic brain injury and could inhibit the apoptosis of damaged cerebral microvascular ECs by activating the AKT signaling pathway. It also inhibited inflammation by blocking the nuclear factor κB (NF-κB) signaling pathway to protect the blood–brain barrier. Meanwhile, studies have found that miR-21-5p is involved in the regulation of cardiomyocytes [26,27]. Hun et al. [28] have found that miR-21-5p acts via the PTEN/Akt/FOXO3a signaling pathway to prevent cardiomyocyte injury caused by high glucose/high fat conditions. Although many studies have investigated the relationship between miR-21-5p expression and ECs, only a few studies have reported on the effect of miR-21-5p expression on EPCs function. In this study, the relationship between miR-21-5p expression and EPCs was examined, and miR-21-5p overexpression was found to significantly improve the proliferation and migration function of EPCs, which are of great significance for the prevention of cardiovascular diseases. miR-21-5p expression can promote angiogenesis and cardiomyocyte survival by participating in the regulation of other signaling pathways [29,30]. Luther et al. [31] found that miR-21-5p expression synergically downregulated the expressions of signaling pathways related to cell apoptosis, thus reducing the death rate of myocardial cells induced by ischemia and playing a protective role for the heart.

In this study, we found that upregulation of miR-21-5p expression could inhibit TSP-1 expression, indicating that miR-21-5p is a key regulator of the proliferation and migration functions of EPCs via TSP-1 expression. Hui et al. [17] also found that upregulation of miR-21-5p expression in ECs can inhibit TSP-1 expression and improve the proliferation, migration, and angiogenesis of ECs.

This study shows that inhibition of TSP-1 expression can reduces the senescence rate of EPCs. Previous studies have found that TSP-1 expression is upregulated in vascular ECs [32,33] and cardiomyocytes of aging individuals [13]. The TSP-1 expression level in cardiomyocytes was significantly upregulated in aged mice (30 months of age) compared with young mice (2 months of age) [14]. An in vitro culture of senescent human fibroblasts in a conditioned medium showed increased expression levels of TSP-1 mRNA and protein, and cell senescence was delayed when cells were cultured with a conditioned medium supplemented with TSP-1 antibody [34]. Lung ECs from young wild-type mice cocultured with TSP-1 also showed a rapid senescent phenotype, with flattening, β-galactosidase activation, and abundant vacuole formation [35]. Exogenous upregulation of TSP-1 expression accelerated the senescence of human lung cells [36] and mouse brain microvascular ECs [33]. Surprisingly, although the cell senescence rate decreased after inhibition of TSP-1 expression, it did not decrease after the inhibition of TSP-1 expression by miR-21-5p mimic, which may be because the degree of inhibition of TSP-1 expression by miR-21-5p mimic was not enough to significantly affect cell senescence.

TSP-1 expression has been shown to inhibit EC proliferation [37,38,39], and a few studies have examined the effect of TSP-1 expression on EPCs function. Tie et al. [40] found that the proliferation function of EPCs was significantly improved after inhibiting TSP-1 expression in type I diabetic mice. Qin et al. [41] measured plasma TSP-1 levels in patients with chronic vascular occlusion and found that the plasma TSP-1 level was higher in patients with collateral circulation disorders of different levels of severity than in patients with good collateral circulation. Thus, high plasma TSP-1 levels could inhibit early and late EPCs angiogenesis. This inhibition may be mediated by the interaction between TSP-1 and CD47, which leads to the downregulation of vascular endothelial growth factor receptor-2 phosphorylation. The interaction between TSP-1 and CD47 may inhibit the replication and growth of progenitor cells. Studies have found that TSP-1 or CD47 expression is absent in the lung ECs of 2- to 3-month-old mice, enabling young mouse cells to maintain their replication and passage functions for >6 months [42]. This suggests that these genes play important roles in the aging process. A new study on umbilical-cord-derived EPCs from preterm infants demonstrated that increased TSP-1 expression level is related to decreased cell replication, while silencing TSP-1 expression can restore cell replication function [43]. Inhibiting the binding of TSP-1 to receptors with blockers will also improve cell proliferation function [35]. In this study, we found that inhibition of TSP-1 expression can significantly improves the proliferation and migration functions of EPCs, which is of great significance for the ability of EPCs to repair damaged vascular endothelium and improve endothelial function. In addition, it can prevent the occurrence of cardiovascular diseases in aging individuals.

## 5. Conclusions

In conclusion, the present study demonstrated for the first time that long-term aerobic exercise can improve the proliferation and migration functions of EPCs in aging individuals by upregulating miR-21-5p expression and inhibiting TSP-1 expression and can reduce the senescence rate of EPCs by downregulating TSP-1 expression. The results of the molecular experiments emphasized that miR-21-5p is a key regulator of the proliferation and migration functions of EPCs via TSP-1 expression. This study provides valuable information for understanding the mechanism by which exercise improves vascular function in aging individuals and for exploring the role of miRNAs in the treatment of EPCs function. However, study limitations cannot be ignored; it is regrettable that no further modeling of vascular endothelial injury has been carried out to examine the function of exercise in the localization of EPCs to the damaged vascular endothelium. We suggest that future experiments should be conducted by modeling to investigate the influence of exercise on the localization and re-endothelialization functions of EPCs.

## Figures and Tables

**Figure 1 ijerph-20-01255-f001:**
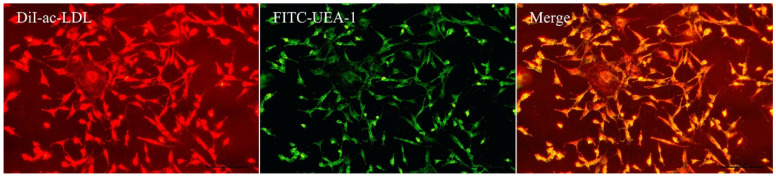
Characterization of endothelial progenitor cells (EPCs). Fluorescent analyses to determine the Dil-ac-LDL uptake and FITC-UEA-1-binding capability of EPCs. Bar, 100 μm.

**Figure 2 ijerph-20-01255-f002:**
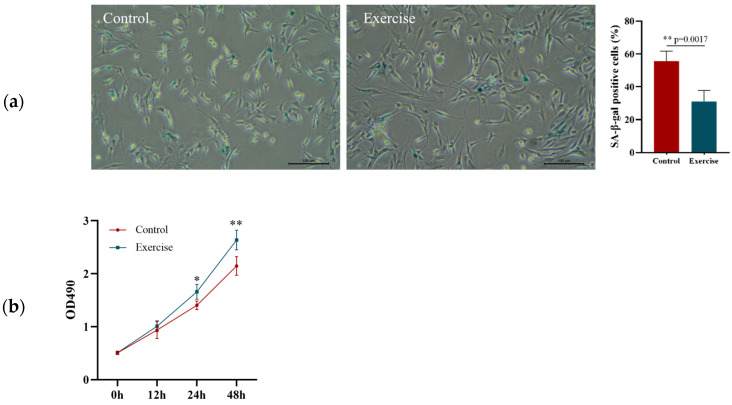
Effect of 8-week aerobic exercise on the senescence, proliferation, and migration of EPCs in aged rats. (**a**) Senescence rate of EPCs determined using SA-β-gal staining. (**b**) Cell proliferation analyzed using MTT assays. (**c**) Cell migration rates at 0 and 24 h after scratching. (**d**) Comparison of miR-21-5p and TSP-1 expression between control and exercise groups. * *p* < 0.05, ** *p* < 0.01. Bar, 100 μm.

**Figure 3 ijerph-20-01255-f003:**
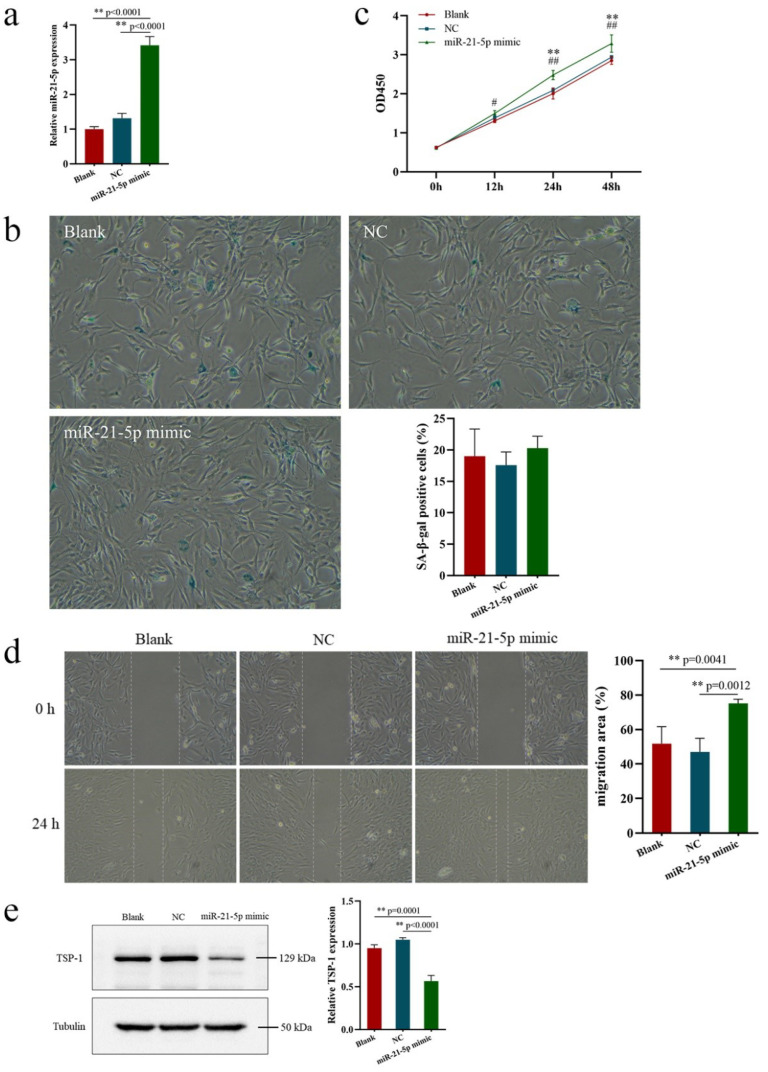
Effects of miR-21-5p overexpression on EPCs function and TSP-1 expression. (**a**) The miR-21-5p expression in EPCs was significantly increased after transfection with mimic for 24 h. (**b**) No significant difference in cell senescence rate after transfection with miR-21-5p mimic for 24 h was found among all the groups. (**c**) Proliferation of EPCs after transfection. (**d**) Migration function of EPCs after transfection. (**e**) The TSP-1 protein expression was significantly lower in the miR-21-5p mimic group than in the NC and blank groups. ^#^ *p* < 0.05, ^##^ *p* < 0.01, ** *p* < 0.01. Bar, 100 μm.

**Figure 4 ijerph-20-01255-f004:**
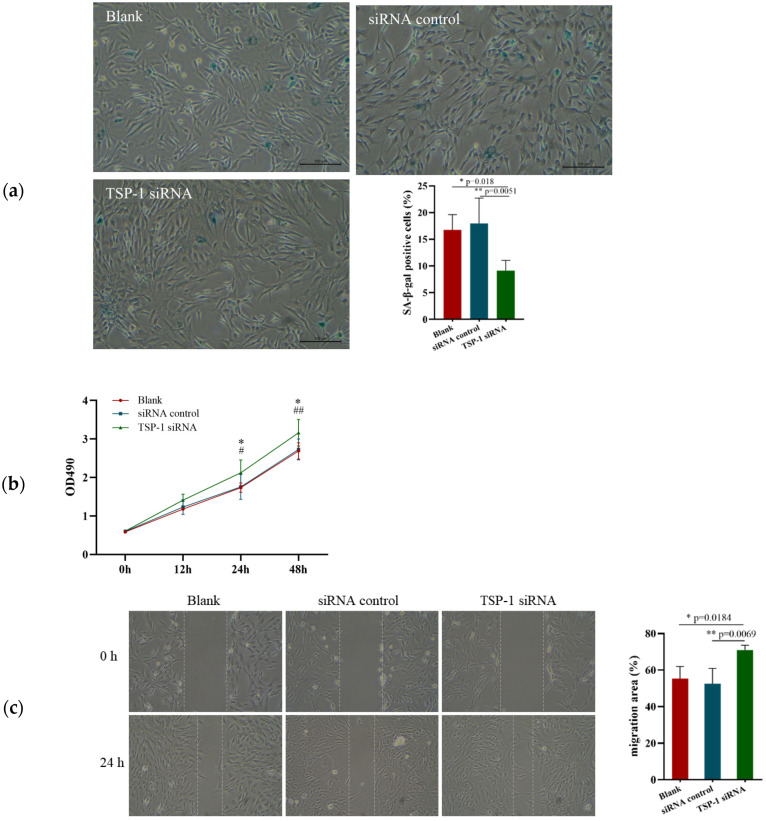
Effects of TSP-1 expression on EPCs function. (**a**) Cell senescence rate after transfection with TSP-1 siRNA for 24 h. (**b**) Proliferation of EPCs after transfection. (**c**). Migration function of EPCs after transfection. ^#^ *p* < 0.05, ^##^ *p* < 0.01, * *p* < 0.05, ** *p* < 0.01. Bar, 100 μm.

**Figure 5 ijerph-20-01255-f005:**
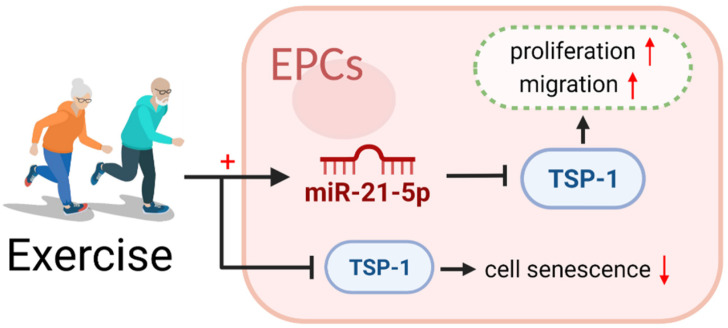
Long-term aerobic exercise improves the proliferation and migration functions of EPCs in aging individuals by upregulating miR-21-5p expression and inhibiting TSP-1 expression. It reduces the senescence rate of EPCs by downregulating TSP-1 expression. miR-21-5p is a key regulator of the proliferation and migration functions of EPCs via TSP-1 expression.

**Table 1 ijerph-20-01255-t001:** Exercise intervention program for 8 weeks.

Week	Mon	Tue	Wed	Thur	Fri
1	15 × 20	15 × 20	15 × 20	15 × 20	15 × 20
2	16 × 30	16 × 30	16 × 30	16 × 30	16 × 30
3	17 × 40	17 × 40	17 × 40	17 × 40	17 × 40
4	18 × 50	18 × 50	18 × 50	18 × 50	18 × 50
5	20 × 50	20 × 50	20 × 50	20 × 50	20 × 50
6	20 × 50	20 × 50	20 × 50	20 × 50	20 × 50
7	22 × 50	22 × 50	22 × 50	22 × 50	22 × 50
8	22 × 50	22 × 50	22 × 50	22 × 50	22 × 50

Unit: m/min × min.

**Table 2 ijerph-20-01255-t002:** Primers used for RNA interference.

Gene	Sequences (5′-3′)
rno-miR-21-5p mimic	Forward: UAGCUUAUCAGACUGAUGUUGA
	Reverse: AACAUCAGUCUGAUAAGCUAUU
rno-mimic NC	Forward: UUCUCCGAACGUGUCACGUTT
	Reverse: ACGUGACACGUUCGGAGAATT
TSP-1-rat-589	Forward: GCAUCUUCACAAGGGAUUUT
	Reverse: AAAUCCCUUGUGAAGAUGCTT
TSP-1-rat-1951	Forward: CCAACAAACAGGUGUGCAATT
	Reverse: UUGCACACCUGUUUGUUGGTT
TSP-1-rat-3317	Forward: GCAUGACCCUCGUCACAUUTT
	Reverse: AAUGUGACGAGGGUCAUGCTT
siRNA control	Forward: UUCUCCGAACGUGUCACGUTT
	Reverse: ACGUGACACGUUCGGAGAATT

## Data Availability

All data generated or analyzed during this study are included in this published article.

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
