# Peer review of "The Mechanism of miR-21-5p/TSP-1-Mediating Exercise on the Function of Endothelial Progenitor Cells in Aged Rats"

_ijerph, 2023, doi:10.3390/ijerph20021255_

Round 1

Reviewer 1 Report

This manuscript discusses exercise can improve the function decline of peripheral endothelial progenitor cells caused by aging, and deeply studies the molecular mechanism of exercise to improve the function of endothelial progenitor cells by miR-21-5p and TSP-1, that is very meaningful. But lack of the pathway concerned, also some mistakes need to be corrected. 

1.     L113,the OD value was measured at 490nm, but in Figure 2c, 3c and 4b, the Y-axis are shown at OD450 and OD45? 

2.     Figure 3b and Figure 4a are lack of the scale. 

3.     This manuscript found that long time aerobic exercise can improve the proliferation and migration functions of EPCs in age rats by upregulating miR-21-5p and inhibiting TSP-1, shown many times in the abstract, results, discussion and conclusion. Readers cannot forget it. So, a simple picture, Figure 5 is necessary or not for such a question?

4. It is a pity that how miR-21-5p or TSP-1 regulate the EPCs, even the function for the age,  the  relate pathway did not show.

Author Response

Comments 1

  1. L113,the OD value was measured at 490nm, but in Figure 2c, 3c and 4b, the Y-axis are shown at OD450 and OD45? 

Response: Thank you very much for your patient review and modification comments, we've corrected the Y-axis.

  1. Figure 3b and Figure 4a are lack of the scale. 

Response: We added the scale.

  1. This manuscript found that long time aerobic exercise can improve the proliferation and migration functions of EPCs in age rats by upregulating miR-21-5p and inhibiting TSP-1, shown many times in the abstract, results, discussion and conclusion. Readers cannot forget it. So, a simple picture, Figure 5 is necessary or not for such a question?

Response: Thank you for your suggestion. We have deeply considered your suggestion, but we considered keeping Figure 5 to better help readers quickly understand the results of this study.

  1. It is a pity that how miR-21-5p or TSP-1 regulate the EPCs, even the function for the age, the relate pathway did not show.

Response: The aim of this study is to explore the function of EPCs regulated by exercise from the perspective of small RNA, and it was also found that exercise can improve the functions of EPCs in aging individuals by downregulating TSP-1 expression via miR-21-5p. The effects of exercise mediated by related pathways on age-related EPCs function will be explored in future studies.

Reviewer 2 Report

In this paper, authors study the Mechanism of miR-21-5p/TSP-1 Mediating Exercise on the Function of Endothelial Progenitor Cells in Aged Rats. Thirty aged (18-month-old) male SpragueDawley rats were randomly divided into control and exercise groups. An aerobic exercise intervention was performed 5 days/week for 8 weeks. EPCs functions, miR-21-5p and TSP-1 expressions were detected after the intervention. The senescence rate, proliferation, and migration of EPCs were examined after overexpressed miR-21-5p and inhibition of TSP-1 expression. Outcomes prove that Aerobic exercise can improve EPCs function in aging individuals via upregulation of miR-21-5p and inhibition of TSP-1, and reduce the EPCs senescence rate by downregulating TSP-1. miR-21-5p is a key regulator of the proliferation and migration functions of EPCs via TSP-1. 

This is an interesting work; however, I have some comments:

- Abstract needs to be rewritten to understand the summary of the work done highlighting the principal contributions.

- In the introduction, the comparison with previous works must be more precise in order to highlight the real contribution of this work. In addition, the motivation and background of wide practical use of the theoretic results presented should be clearly emphasized to facilitate the readers.

- Some figures are not clear. Please provide a high-resolution figure and check the selected colors.

- Please insert some recent references. 

- Conclusion does not reflect the presented work. It should be rewritten by adding the limitations and perspectives.

Concluding, the paper has potential to be appreciated by the readers and the above comment are formulated such that to enhance its impact.

Author Response

Comments 2

  1. Abstract needs to be rewritten to understand the summary of the work done highlighting the principal contributions.

Response: Thank you for your valuable suggestion. We have rewritten the abstract according to your suggestion. The principal contributions are highlighted by adding: “Conclusions: Our results indicate that long-term aerobic exercise can improve the functions of EPCs in aging individuals by downregulating TSP-1 expression via miR-21-5p, which reveals the mechanism of exercise in improving cardiovascular function”. (See line 7-22)

  1. In the introduction, the comparison with previous works must be more precise in order to highlight the real contribution of this work. In addition, the motivation and background of wide practical use of the theoretic results presented should be clearly emphasized to facilitate the readers.

Response: We appreciate this kind suggestion and have added more relevant information. (See line 57-67)

  1. Some figures are not clear. Please provide a high-resolution figure and check the selected colors.

Response: We have provided high-resolution figure.

  1. Please insert some recent references. 

Response: We have reviewed new literature published in the last two years and inserted recent references relevant to the topic of this study. (See line 242 and 250-252)

  1. Conclusion does not reflect the presented work. It should be rewritten by adding the limitations and perspectives.

Response: We have taken your advice and added relevant information in the part of conclusion, the limitations and perspectives added is: “However, study limitations cannot be ignored, it is regrettable that no further modeling of vascular endothelial injury has been carried out to examine the function of exercise in the localization of EPCs to the damaged vascular endothelium. We suggest that future experiments should be conducted by modeling to investigate the influence of exercise on the localization and reendothelialization functions of EPCs.” (See line 305-308)

Round 2

Reviewer 2 Report

All my suggestions have been implemented in the paper, I have no further comments. 

Best regards!